# ImpactETC1.0: Impact-oriented tracking of extratropical cyclones with global optimisation and track reconciliation

Niels Agertoft<sup>1</sup>, Jian Su<sup>1</sup>, Jonas Wied Pedersen<sup>1,2</sup>, Ida Margrethe Ringgaard<sup>1</sup>, and Morten Andreas Dahl Larsen<sup>1</sup>

Correspondence: Niels Agertoft (nie@dmi.dk)

**Abstract.** Extratropical cyclones (ETCs) play a critical role in shaping extreme weather events in the Nordic region, often driving storm surges, heavy precipitation, and high winds that can lead to significant socio-economic and environmental impacts. However, traditional cyclone tracking methods focus primarily on large-scale atmospheric dynamics without explicitly linking cyclone characteristics to their regional impacts. To address this gap, we introduce ImpactETC1.0, a novel framework designed to identify and track ETCs with a specific focus on their impacts, here illustrated for the case of storm surges. The framework includes several novel algorithmic features, including global optimisation for the correspondence problem, BLOB analysis techniques for track fragmentation issues arising from surface-level tracking over complex terrain, and automated calibration of post-processing parameters. Applied to the CERRA reanalysis dataset and with a focus on the Nordic region, ImpactETC1.0 successfully reconstructed ETC tracks across complex terrain and during periods of rapid storm evolution, while keeping computational costs low. Compared with a standard nearest-neighbour heuristics, the global optimisation reduced suboptimal connections by up to one-third, at negligible additional runtime. The track reconciliation step was essential in preventing track fragmentation and premature termination of tracks, producing storm tracks that were, on average, twice as long over complex land-ocean boundaries and mountain ranges. As the post-processing step is extremely quick to perform, a sensitivity analysis could be done, and a score named the Single Storm Score for automated calibration of filtering parameters was developed. A key strength of ImpactETC1.0 is the use of global optimisation and track reconciliation, as the need for these in storm tracking will only grow with the increasing resolution of data sets. Together, these results demonstrate that ImpactETC1.0 enables accurate and impact-relevant ETC tracking.

## 1 Introduction

The tracking of extratropical cyclones (ETCs) has been a longstanding challenge in meteorology, with various methodologies developed to identify and analyse their evolution (see Walker et al. (2020), and references therein). Conventional ETC tracking methods rely on objective feature tracking algorithms that detect cyclones using observed or simulated atmospheric datasets (Hodges, 1995; Hodges et al., 2003; Neu et al., 2013; Raible et al., 2008; Walker et al., 2020). These methods have been widely applied in climate studies to analyse storm climatology and trends (Bengtsson et al., 2009; Feser et al., 2015; Hoskins and

<sup>&</sup>lt;sup>1</sup>Danish Meteorological Institute, Sankt Kjelds Plads 11, Copenhagen, 2100, Denmark

<sup>&</sup>lt;sup>2</sup>DTU Sustain, Technical University of Denmark, Bygningstorvet, Building 115, Kgs. Lyngby, 2800, Denmark

Hodges, 2019; Lodise et al., 2022), they are also increasingly used in operational weather forecasting (Froude, 2010), extreme weather detection (Ullrich et al., 2021), wind energy risk evaluation (Gonçalves et al., 2021), and risk impact assessment (Hunter et al., 2016).

Tracking frameworks traditionally adapt the conventional three-step storm tracking process: identification, tracking, and post-processing (Neu et al., 2013; Walker et al., 2020). Each of these steps introduces methodological decisions that influence the final tracks. In the identification step, ETCs have to be geographically located in each individual time step, which is usually done by analysing atmospheric fields and identifying ETC characteristics such as pressure minima or rotational movements around their centre. The next step is to track ETCs through time by solving the so-called "correspondence problem". This is done by linking identified ETC locations across time steps to form coherent tracks. The final post-processing step typically aims to filter out irrelevant tracks and sometimes smoothen noisy or abrupt ETC movement. Differences in algorithm design and the choice of atmospheric dataset have resulted in significant variability in cyclone track statistics across different studies, making direct comparisons challenging (Feser et al., 2015; Flaounas et al., 2023; Neu et al., 2013; Walker et al., 2020). Comparison studies show that different tracking algorithms, even when applied to the same dataset, produce cyclone counts that differ significantly, highlighting the sensitivity to identification criteria, correspondence strategies, and post-processing filters (Grieger et al., 2018; Neu et al., 2013; Raible et al., 2008). These discrepancies are particularly pronounced in regions with complex terrain (Medina and Houze Jr., 2016).

Many conventional tracking algorithms prioritise identifying storm centres and tracking their movement over time, without explicitly linking cyclone characteristics to their impact-related consequences, such as flooding, storm surges, and wind damage (Walker et al., 2020). This is reflected in the ETC identification step, where conventional definitions of ETCs often rely solely on dynamical thresholds such as minimum mean sea level pressure (MSLP), and in the post-processing filter parameters, which are defined without validating whether the resulting tracks correspond to relevant observed surface impacts (Zappa et al., 2013; Catto, 2016).

The choice of atmospheric variable used in the ETC identification step strongly influences the resulting tracks (Walker et al., 2020). Common choices are MSLP and its derivatives (Lodise et al., 2022; Peréz-Alarcón et al., 2024; Ragone et al., 2018; Ullrich et al., 2021), relative vorticity (Flaounas et al., 2014; Lakkis et al., 2019; Hoskins and Hodges, 2019), and geopotential height at various pressure levels (Aragão and Porcù, 2022; Hofstätter et al., 2016). Because the most severe impacts of ETCs occur at the surface, tracking the centre of the cyclone at or near the surface is desirable from an impact-focused perspective. However, surface-level variables can present significant challenges in regions with complex terrain. MSLP is not directly modelled in reanalysis data, but instead derived from surface pressure via a hydrostatic approximation, which often breaks down over land, especially in mountainous areas. Similarly, near-surface relative vorticity is affected by strong terrain gradients and land-ocean boundaries. An alternative is to track ETCs using upper-level variables such as geopotential height or relative vorticity at levels of 850 hPa or 700 hPa (Flaounas et al., 2014; Hofstätter et al., 2016; Priestley et al., 2020; Sanchez-Gomez and Somot, 2018). However, the position and evolution of the upper-level features can diverge from the cyclone centre at the surface, and some ETC's are shallow, making them difficult or impossible to track higher in the atmosphere (Lakkis et al., 2019). This divergence can potentially cause a misrepresentation of the storm's surface impacts. Hence, there is a need for


methods that retain the impact-relevance of surface-level tracking while addressing the artefacts and discontinuities that arise near complex terrain.

Regarding the correspondence problem, most existing algorithms have implemented variations of a greedy nearest-neighbour heuristic where the closest available points in a sequence of time steps are connected (e.g., Aragão and Porcù, 2022; Flaounas et al., 2014; Peréz-Alarcón et al., 2024; Priestley et al., 2020; Ullrich et al., 2021). While nearest-neighbour heuristics are computationally efficient, they may lead to implausible or suboptimal connections, especially in dense storm fields or large, vague low-pressure areas. In such cases, there can be multiple possible ETC centres to connect to within a defined neighbourhood region. Some algorithms simply choose the nearest neighbouring point (Aragão and Porcù, 2022; Ullrich et al., 2021), while others make a prediction of where they expect the next point to appear and choose the options closest to that (Hofstätter et al., 2016; Sanchez-Gomez and Somot, 2018). Other algorithms make the choice dependent on the intensity of the ETC centre in the identification variable field and choose the option with the strongest extrema (Lodise et al., 2022; Peréz-Alarcón et al., 2024), or the option that gives the least difference in value between time steps (Flaounas et al., 2014). It is common to deny connections above a certain distance (Aragão and Porcù, 2022; Lodise et al., 2022; Ragone et al., 2018), or if the difference exceeds a predefined threshold (Sanchez-Gomez and Somot, 2018). Some algorithms allow for a small gap in the tracking procedure if it is not possible to locate the ETC centre for a short period of time to avoid the track breaking (Peréz-Alarcón et al., 2024; Ullrich et al., 2021). These additional features increase the likelihood of making good connections, but do not guarantee that the correct connection is made. Moreover, in each individual correspondence problem, this issue gradually worsens as the connection possibilities are exhausted, further decreasing the likelihood of finding the optimal match.

Post-processing steps typically aim to filter out identified ETCs that are deemed insignificant to a particular study. This can be short-lived systems filtered with parameters such as minimum storm duration and minimum track length (Aragão and Porcù, 2022; Lodise et al., 2022; Ullrich et al., 2021), or systems deemed too low intensity with thresholds on maximum MSLP or minimum relative vorticity along tracks (Ragone et al., 2018; Sanchez-Gomez and Somot, 2018). Parameter values for these post-processing filters are rarely calibrated against impact data. Instead, they are determined ad hoc for a given study purpose, often simply through round values such as a minimum duration of 24 hours and a minimum length of 1000 km. As a result, these methods may retain dynamically strong but impact-irrelevant systems or, conversely, discard slower-moving, small, or short-lived storms that drive local extremes. The lack of a consistent, impact-oriented filtering approach presents a major limitation to applying storm track analysis to hazard assessment or long-term climate adaptation planning.

Motivated by these challenges, we introduce a novel ETC tracking framework designed to enhance the relevance of ETC tracks for on-the-ground impact assessments. The new framework contains several scientific novelties:

1. A global, yet computationally fast, optimisation solution for the correspondence problem with the Hungarian Algorithm (Kuhn, 1955; Munkres, 1957). This includes benchmarking against a typical nearest-neighbour heuristics to assess tradeoffs between computational efficiency and tracking continuity.


- 2. Binary Large Object (BLOB) analysis techniques to address discontinuous jumps and fragmentation of storm tracks caused by MSLP issues, thereby improving surface-level tracking of the ETC centre over complex terrain and land-sea boundaries.
- 3. Impact-oriented calibration of common post-processing parameters, such as minimum track length and duration, as well as spatial proximity to the impact location. This includes an assessment of the ability of the algorithm to capture the correct number of ETCs per impact event.

The ETC tracking framework is applied and evaluated for historical (1991-2020) storm surge events in Denmark, which constitutes a complex case due to the diverse nature of storms and ocean dynamics across the Atlantic-North Sea-Baltic Sea trajectory (Andrée et al., 2021, 2022; Andrée et al., 2023).

## 100 2 Data and region of study

The CERRA dataset used in this study is a high-resolution regional reanalysis product developed by the Copernicus Climate Change Service (C3S) to support climate applications in Europe (Ridal et al., 2024). CERRA has a spatial resolution of 5.5 km and an hourly temporal resolution, making it particularly well suited to capture mesoscale features of ETCs across complex European terrains (see Figure 1). The dataset assimilates a wide range of observational inputs and is dynamically downscaled from the ERA5 global reanalysis using the HARMONIE-AROME model configuration. For storm tracking purposes, we employ two key atmospheric parameters: MSLP and relative vorticity at 500 hPa,  $\zeta_{500}$ . Other tracking algorithms typically opt to use  $\zeta$  at 850 hPa or 700 hPa, but our experience with the study region was that the Scandinavian mountains were causing artefacts at these levels. Before calculating  $\zeta_{500}$ , all variables were re-gridded from the native CERRA grid to a regular latitude—longitude grid using bilinear interpolation to simplify subsequent processing and ensure consistency in spatial derivatives. MSLP is used to identify the low-pressure centres that are typically associated with ETC cores.  $\zeta_{500}$  was calculated from the re-gridded wind components u and v using a centred finite-difference scheme,

$$\zeta = \frac{\partial v}{\partial x} - \frac{\partial u}{\partial y}$$



To identify and evaluate impact-relevant storm tracks, a set of significant storm surge events was compiled based on observed water level time series from Danish tide gauge stations (see locations in Figure 1). Events were selected using an empirical threshold corresponding to the 5-year return level at each station, ensuring that only significant extremes were included. To avoid double counting of close extremes, a minimum time interval of 1.5 days between two events was imposed for both the same station and for events that occur across several stations. This selection process produced a set of dates in which coastal water levels exceeded thresholds indicative of high-impact events. We also performed a manual review of each case to omit erroneous recordings and to ensure that the elevated water levels were driven by extratropical cyclone activity rather than other mechanisms such as seiches or large anticyclones. As a result of this, six dates were excluded from the analysis due to the absence of an identifiable ETC driver. For all events used, the time of impact was defined based on the peak in the water level time series, which provides a physically consistent reference point for initialising the storm tracking algorithm.

**Figure 1.** A map of Denmark with the locations of the water level stations used to identify significant impact events as red points (left). The full CERRA domain with surface topography (colour scale) and ocean mask (blue) as defined in the CERRA data set as well as a red line denoting the "area of relevance" covering the North Sea-Baltic Sea region (right).

## 3 ETC tracking algorithm

The ETC detection and tracking algorithm of this paper begins with the user specifying a "time of impact", e.g. the date and hour of a local storm surge event. The algorithm then searches for and tracks a potential ETC within a user-defined time window around the time of impact. Figure 2 presents a schematic overview of the core components within the ImpactETC1.0 tracking framework. The method is designed to systematically identify and isolate extratropical cyclones responsible for local surface impacts within a specified time window. Each stage in the workflow builds on the previous one, progressively refining the storm tracks from initial detection to final filtering. The figure outlines the logical flow between components, from storm centre identification to track linkage, spatial consolidation, and impact-based filtering. The sections are inherently independent in nature and thus can be seamlessly swapped for alternative approaches. The following subsections (3.1-3.4) detail the implementation and rationale behind each step, while subsection (3.5) provides an overview of the parameters involved.

## ImpactETC - ETC Tracking Framework

Figure 2. Overview of the ImpactETC1.0 framework for identifying ETCs linked to observed surface impacts. The top panels illustrate the motivation using the Storm Bodil case (December 2013): (left) flooding in Frederikssund, Denmark (Photo: Martin Stendel), (centre) detided water level at a coastal station with the  $\pm 24$  h impact window shaded, and (right) the associated ETC seen in satellite imagery. These impacts motivate the bottom panel's algorithm, which proceeds in four main stages: (1) Candidate Point Identification—cyclone centres are located by detecting MSLP minima and filtering using vorticity and spatial thresholds; (2) Solving the Correspondence Problem—storm centres are linked across time using either a nearest-neighbour heuristic or the Hungarian Algorithm, with constraints on distance and cost; (3) BLOB Analysis—track discontinuities caused by terrain or domain boundaries are resolved by reconnecting nearby low-pressure systems; (4) Post-Processing—tracks are retained if they satisfy criteria for duration, travel distance, and overlap with a defined "area of relevance", with parameters tuned through grid search. The final output in the top right panel consists of impact-relevant ETC tracks capable of explaining localised storm surge or flooding events.

## 3.1 Identifying candidate points

Since the aim is to identify the relevant ETC that caused a local impact at a specific time, we first define a time window around the time of impact with start and end time stamps of  $t_s$  and  $t_e$ , respectively. For the application of the algorithm to storm surge events in this paper, a time window of  $\pm$  24 hours was chosen around the beginning peak and the end peak of the storm surge event, which constitutes a minimum of 49 time steps per event. This peak is highlighted in figure 3.




Figure 3. Time series of detided water level at Ribe, Denmark, during a storm surge event in January 2005. The dashed red line marks the time of maximum water level. The shaded region represents the  $\pm 24$ -hour window around the peak used to identify and track the associated ETC.

Within this time window, the algorithm identifies candidate locations for the centre of an ETC by searching for local MSLP minima in the entire CERRA domain (Ridal et al., 2024). Across large spatial domains with high resolution grids, there are two issues: 1) the number of grid cells is large and simple local neighbourhood searches will generate multiple local minima, and 2) there may be multiple ETC's present within the domain at any given time.

To solve these issues, we first attempted using a kernel around each point to determine its status as local minima, but this approach proved too computationally expensive. Instead, the goal is to upscale the original grid into coarser cells without missing any local minima due to overly large cells. To ensure this, we require that two potential local minima located at opposite corners in the upscaled grid cell are no further apart than a pruning radius r. Using the Pythagorean theorem, we can derive the length of the squares from the diagonal, given by the pruning radius r as the following:  $l = \sqrt{\frac{r^2}{2}}$ . We proceed through the following steps (Figure 4 shows the remaining potential candidate points after each step):

- 1. Determine the local MSLP minimum within each upscaled cell (Figure 4 a).
- 2. Remove local MSLP minima if any neighbouring upscaled cells within the pruning radius r contain a lower MSLP minimum (Figure 4b).
  - 3. Remove local MSLP minima with values above "MSLP $_{max}$ ", since these are deemed too high to be an ETC centre (Figure 4 c).
  - 4. Remove local minima that do not have a  $\zeta_{500}$  intensity above  $\zeta_{500\,min}$  within a radius of 500 km (Figure 4 d).

From these steps, we identify potential candidate points that represent the centre of an ETC. These will be used in the correspondence problem in the next step of the algorithm.





The hyperparameters in these steps  $(r, \text{maximum MSLP value at ETC centre}, \text{ and minimum regional } \zeta_{500}$  intensity) can all be fine-tuned for a given application of the algorithm. For the storm surge examples in this paper, we use the values  $r = 350 \, \text{km}$ ,  $\zeta_{500 \, min} = 1.5 \cdot 10^{-4} \, \text{s}^{-1}$ , and  $\text{MSLP}_{max} = 1010 \, \text{hPa}$ . The choice of r can have a large impact on the computational efficiency of the algorithm, which will be investigated further in Section 5.2. In general, lax pruning constraints will lead to more candidate points, which in turn impacts the computational efficiency of the algorithm.

#### 3.2 Solving the correspondence problem

The candidate points found in section 3.1 for each individual time step must now be connected through adjacent time steps to determine the storm tracks - a task known as the correspondence problem in the storm tracking literature (Walker et al., 2020). The general objective is to minimise the overall distance between the pairs of connected candidate points at time  $t_i$  and  $t_{i+1}$ . An added complication is that the number of candidate points in each time step is not guaranteed to be identical. If we define n and m as the number of candidate points at time  $t_i$  and  $t_{i+1}$ , respectively, then the problem can be either balanced if n = m or unbalanced if  $n \neq m$ . A theoretical solution to this problem can be found through global optimisation. However, due to concerns about the computational requirements of global optimisation, it is common in current storm tracking algorithms to use a heuristic approach. This is often based on the nearest-neighbour approach, where one iterates through the candidate points at  $t_i$  and sequentially connects them to the candidate point closest in space at  $t_{i+1}$ . Usually, a threshold for the maximum possible travel distance of a storm centre between two time steps,  $D_{max}$ , is part of the heuristic algorithms. Heuristic-based approaches are not guaranteed to provide the correct solution to the problem but have the benefit of being very fast to compute.

In our tracking framework, we have implemented a global optimisation solution that avoids some of the pitfalls from nearest-neighbour heuristics. This type of correspondence problem is formally known as the "assignment problem" in optimisation theory, and it can be efficiently solved using the Hungarian Algorithm. The algorithm is well studied in other fields, such as logistics (Seda, 2022) and scheduling (Zhang et al., 2024), and therefore we will only provide a general description of it here. The Hungarian algorithm begins with setting up a cost matrix, which here will be an  $n \times m$  distance matrix, which represents the distance between all n candidate points at  $t_i$  and all m points at  $t_{i+1}$ . Equation 1 exemplifies how such a distance matrix is set up, with the leftmost matrix in Equation 1 being  $n \times m$ . The algorithm requires that the distance matrix is square, and since that is not guaranteed, the next step is to pad the matrix with phantom candidate points that have a maximum cost associated with them (middle matrix in Equation 1). We then apply a maximum allowable distance for a connection to be "realistic",  $D_{max}$ , which in the storm surge implementation of this paper is set to 600 km. This maximum distance is implemented by censoring all matrix elements with values above  $D_{max}$  (see rightmost matrix in Equation 1).

$$\begin{bmatrix} 2588 & 105 & 83 \\ 6 & 2640 & 2510 \\ 2395 & 254 & 112 \\ 319 & 2694 & 2577 \end{bmatrix} \rightarrow \begin{bmatrix} 2588 & 105 & 83 & D_{max} \\ 6 & 2640 & 2510 & D_{max} \\ 2395 & 254 & 112 & D_{max} \\ 319 & 2694 & 2577 & D_{max} \end{bmatrix} \rightarrow \begin{bmatrix} D_{max} & 105 & 83 & D_{max} \\ 6 & D_{max} & D_{max} & D_{max} \\ D_{max} & 254 & 112 & D_{max} \\ 319 & D_{max} & D_{max} & D_{max} \end{bmatrix}$$
(1)

Figure 4. Visualization of the progressive filtering of candidate cyclone centres using MSLP and 500 hPa vorticity criteria. From left to right, top to bottom: all local MSLP minima identified within upscaled grid cells. After applying radius-based culling, retaining only the lowest minimum within a defined neighbourhood. After removing candidates with MSLP values above the threshold MSLP $_{max}$ . After further filtering out candidates lacking sufficient mid-tropospheric vorticity within a 500 km radius. The background shading depicts the MSLP field for context. Event is from 1997-10-11.



Once the distance matrix is set up, made square and censored, the algorithm finds the optimal solution by minimising the sum of the distances between all connections. This is done through row/column reduction, covering zeros with lines, and matrix adjustments in the following steps:

- 1. Identify the smallest element in the matrix and subtract it from all elements, thus obtaining a matrix with at least one zero.
- 2. For horizontal and vertical lines through the rows and columns of the matrix, identify the minimum number of lines that would be required to cover all zeros.
  - 3. If the number of lines is exactly the size of the matrix, then an optimal solution has been found. If it is smaller than the size of the matrix, an adjustment to the matrix is needed. The adjustment is to identify the smallest element *h* not covered by a line. Add *h* to all elements covered twice at the intersections of two lines and subtract *h* from all uncovered elements.
  - 4. Repeat steps 2 and 3 until the number of lines required to cover the zeros is equal to the size of the matrix.
  - 5. Select a set of connections that all have zeros. This is the optimal solution.

The connections found by the Hungarian Algorithm represent the solution that minimises the total sum of the connections between  $t_n$  and  $t_{n+1}$ . Computationally, in terms of big-O notation, it scales at  $O(n^3)$ . If multiple optimal solutions with the same objective value exist, the algorithm will return only one of them. For the case of storm tracking with the costs defined as exact geographical distance between points, we deem it unlikely that this will occur.

## 3.3 BLOB analysis for resolving storm track fragmentation over complex terrain

In this study, MSLP is used to identify candidate points for the centres of ETCs. However, MSLP is a diagnostic variable, computed from the modelled surface pressure by estimating the hypothetical sea level pressure, assuming hydrostatic balance. Over complex terrain, this assumption often breaks down, e.g., over the Scandinavian mountains. As a result, artefacts and inaccuracies can arise in the MSLP field. One recurring issue arises when an ETC crosses a mountain range: Candidate centres often appear to "stall" on the upstream side of the mountains, while a new candidate centre simultaneously appears downstream in the same time step. This leads to a discontinuous jump in the storm track, potentially causing the track to "break", resulting in a single ETC being misidentified as two separate systems. To address this, we introduce a novel track reconciliation step using a BLOB analysis technique, which performs synoptic-scale spatial continuity checks based on the MSLP field.

We observed that when such breaks occur, the upstream and downstream candidate points typically remain part of the same synoptic-scale low-pressure system, i.e. in close spatial proximity and with similar MSLP values. BLOB analysis leverages this by identifying contiguous low-pressure "regions" within a given MSLP range.

The method works as follows: for a given candidate point, we define a pressure range (e.g.  $\pm$  5 hPa around its MSLP value).

5 The entire MSLP field is then binarized, with grid points within the range set to 1 (being inside a BLOB), and those outside

220

225

set to 0. The result is BLOB's that represent spatially coherent low-pressure areas. For each time step, we check whether other candidate points fall within the same BLOB as the one containing the current candidate point. (see Figure 5). In certain edge cases, the BLOB's can become very large. To avoid false positives, we impose a maximum allowable BLOB bounding box of 3000 km, along with a threshold on the maximum distance between two candidate points of 2500 km for them to be considered connected.

In simple cases, the discontinuity occurs at the end of one track and the start of the next, which can then be merged. In more complex cases, two tracks may overlap temporally: the upstream track continues for a few time steps while the downstream track has already begun. The task then becomes to choose if and when the jump from the first track to the start of the second track should occur. In these instances, we identify the first time step in which the two candidate points fall within the same BLOB. From that point onwards, we monitor both the continuations of the tracks and select the one that persists for the longest number of time steps as the final coherent track (see Figure 6).

In the results section, we demonstrate the value of this track reconciliation step by showing how frequently it is used on the storm surge events of this study, how it impacts the length of the final tracks, and how it avoids broken tracks being filtered out in the post-processing steps described in the next section.

**Figure 5.** Illustration of the BLOB analysis step used for track reconciliation. (a) This panel shows an ETC crossing the Scandinavian mountains, where the MSLP field causes artificial track fragmentation. (b) The BLOB analysis identifies spatially coherent low-pressure regions (BLOB's) and identifies candidate points within the same BLOB, reconnecting candidate centres upstream and downstream of the mountains to maintain a continuous storm track.

Figure 6. Example of track reconciliation between two fragmented tracks. The points show MSLP minima for each time step indicated by numbers. At time step 16, the algorithm identifies the start of a new track (first white point). From the detection of a BLOB made with a  $\pm$  5 hPa range around the MSLP of the candidate point, it is found that since both the original (black) and new (white) points in time step 16 are within the threshold distance of the BLOB, the algorithm jumps from the black to white track, since in this case, the white track has a longer continuation, as it continues to time step 50, whereas the black track only continues to time step 17. The black x's denote other MSLP minima in the field but since they are not within the same BLOB, they are ruled out as potential connections.

## 230 3.4 Post-processing of tracks

Within the impact time window, multiple ETC tracks are often present throughout the data domain. These tracks vary in length, duration, and spatial trajectory. Therefore, a crucial final step in any storm tracking algorithm is the filtering of tracks that are either insignificant or unrelated to the local impact.

The presented algorithm applies four criteria that an ETC track must satisfy in order to be considered relevant to the impact:

- Minimum travel distance: The ETC must travel a minimum distance, measured as the total length of the track in kilometres.
  - Minimum duration: The ETC must persist for a minimum duration, defined as the time between the first and last detected points along its track.
  - Area of Relevance (AoR): The storm track must enter a predefined spatial area around the impact location.
- Minimum time in AoR: The ETC must spend a minimum amount of time within the AoR.

The first two criteria primarily eliminate short-lived or spatially limited tracks across the domain, while the latter two are designed to filter out tracks that are unlikely to have contributed to the local impact. All four criteria are configurable, underscoring the need for parameter optimisation in the post-processing stage.

## 3.5 Overview of parameters

The full tracking algorithm contains many parameters and hyperparameters, some of which can be reasonably selected from knowledge of the physical system, while others need to be tuned for a given case area. In the literature on tracking algorithms, we often find a lack of (hyper-)parameter documentation, so here we present Table 1, which provides a full and transparent overview of the parameters in this algorithm.

#### 4 Results

250

# 4.1 Correspondence problem: global optimization vs heuristics

Here we present results on the trade-off between solution quality and computational efficiency of the HA algorithm presented in Section 3.2 over a range of complexities of correspondence problems. As part of this, HA is benchmarked against a nearest-neighbour (NN) heuristic. For the NN approach, we employ a greedy algorithm, which iteratively selects the valid minimum distance connection until all pairings are exhausted or exceed a predefined distance threshold  $D_{\text{max}}$  (Algorithm 1):

To explore the influence of problem complexity on NN and HA performance, we vary the pruning radius r used in the candidate point identification step. Smaller r-values yield denser point clouds and larger correspondence problems. We tested six radii: 100, 175, 250, 350, 500, and 700 km, evaluating the run time and three additional metrics that explore potential suboptimal artefacts of the greedy NN solution:

**Table 1.** Overview of parameters in the full ETC tracking algorithm including explanation and the values used in the final setup for the case application.

| Parameter                         | Algorithm Step      | Explanation                                                | Value for this case                |
|-----------------------------------|---------------------|------------------------------------------------------------|------------------------------------|
| Impact window size                | Identify candidates | Time window around the start $(t_s)$ and end $(t_e)$ of    | $\pm 24\mathrm{h}$                 |
|                                   |                     | peaks for that event.                                      |                                    |
| $t_s$                             | Identify candidates | Start date of event                                        | -                                  |
| $t_e$                             | Identify candidates | End date of event                                          | -                                  |
| Pruning radius                    | Identify candidates | A minima is pruned if another minima with lower            | 350 km                             |
|                                   |                     | pressure is found within a given range                     |                                    |
| $MSLP_{max}$                      | Identify candidates | A minima is pruned if it has a pressure above              | 1010 hPa                           |
|                                   |                     | $MSLP_{max}$                                               |                                    |
| $\zeta_{500min}$                  | Identify candidates | A minima is pruned if it exceeds a $\zeta_{500}$ vorticity | $1.5 \cdot 10^{-4} \text{ s}^{-1}$ |
|                                   |                     | threshold within the $\zeta_{500min}$ Range                |                                    |
| $\zeta_{500min}$ Range            | Identify candidates | Allowed range for a minima to associate vorticity.         | 500 km                             |
| $D_{max}$                         | Correspondance      | Maximum allowed distance for connection be-                | 600 km                             |
|                                   |                     | tween minimas in adjacent timesteps.                       |                                    |
| Pressure range                    | BLOB analysis       | Pressure range surrounding the local minima for            | $\pm$ 5 hPa                        |
|                                   |                     | thresholding the MSLP slice prior to BLOB detec-           |                                    |
|                                   |                     | tion                                                       |                                    |
| Max BLOB bounding box             | BLOB analysis       | Maximum height or length of the bounding box               | 3000 km                            |
|                                   |                     | surrounding the identified BLOB                            |                                    |
| Max BLOB candidate point distance | BLOB analysis       | Maximum distance between two points found to be            | 2500 km                            |
|                                   |                     | in the same BLOB                                           |                                    |
| Minimum travel distance           | Post-processing     | Minimum distance an ETC must travel to not be              | 200 km                             |
|                                   |                     | filtered out                                               |                                    |
| Minimum duration                  | Post-processing     | Minimum duration of an ETC for it to not be fil-           | 32 h                               |
|                                   |                     | tered out                                                  |                                    |
| AoR Latitude Min                  | Post-processing     | Minimum latitude of the AoR                                | 50°                                |
| AoR Latitude Max                  | Post-processing     | Maximum latitude of the AoR                                | 70°                                |
| AoR Longitude Min                 | Post-processing     | Minimum longitude of the AoR                               | 0°                                 |
| AoR Longitude Max                 | Post-processing     | Maximum longitude of the AoR                               | 30°                                |
| $\Delta$ AoR                      | Post-processing     | Change in size of the AoR that an ETC must enter           | 0°                                 |
| Minimum time in AoR               | Post-processing     | Minimum amount of time steps spent inside the              | 12 h                               |
|                                   |                     | AoR                                                        |                                    |
| Domain Resolution                 | -                   | Lateral Resolution of the dataset                          | 5.5 km                             |

## Algorithm 1 Nearest neighbour heuristic for the correspondence problem

**Require:** Distance matrix  $\mathbf{D} \in \mathbb{R}^{n \times m}$ , maximum movement distance  $D_{\max}$ 

**Ensure:** Assignment list A

1:  $\mathcal{A} \leftarrow \emptyset$ 

2: while minimum value in **D** is less than  $D_{\text{max}}$  do

3: Find  $(i, j) = \arg\min \mathbf{D}$ 

4: Add (i,j) to A

5: Remove row i and column j from  $\mathbf{D}$ 

6: end while

7: return A

260

- 1. Number of correspondence problem solutions that yield suboptimal matches.
- 2. Number of connections missed relative to HA.
  - 3. Mean difference in the objective value of the correspondence problem, when the NN algorithm differs from HA (in %).

Based on this analysis of the performance of HA and NN for the 37 impact events, a total of 1966 correspondence problems are solved, as summarised in (Table 2). Dates of the impact events are included in Table A1 in the Appendix.

**Table 2.** Comparison of Hungarian Algorithm (HA) and Nearest Neighbor (NN) for different pruning radii. The presented run times are to solve all 1966 correspondance problems across all 37 events.

| Pruning     | Median       | Run time for HA [s]  | Run time for NN [s]  | # Suboptimal NN | NN's # missed   | NN's mean |
|-------------|--------------|----------------------|----------------------|-----------------|-----------------|-----------|
| radius [km] | problem size | Run time for TIA [8] | Run time for two [8] | solutions (%)   | connections (%) | error [%] |
| 700         | 9            | 0.17                 | 0.24                 | 1 (0.05)        | 0 (0.0)         | 1.5       |
| 500         | 12           | 0.19                 | 0.24                 | 0 (0.0)         | 0 (0.0)         | 0.0       |
| 350         | 16           | 0.29                 | 0.33                 | 6 (0.31)        | 0 (0.0)         | 1.1       |
| 250         | 21           | 0.49                 | 0.44                 | 24 (1.22)       | 7 (0.36)        | 6.4       |
| 175         | 29           | 1.2                  | 0.57                 | 99 (5.04)       | 21 (1.07)       | 4.2       |
| 100         | 52           | 8.9                  | 0.80                 | 658 (33.5)      | 126 (6.41)      | 3.2       |

As the radius decreases, the number of candidate points and assignment options increases, which leads to greater problem complexity. Below a radius of 350 km, the NN heuristic shows a sharp increase in suboptimal matches and missed connections. At a 100 km radius, 33.5 % of correspondence problems yield inferior solutions using the NN method, and 6.41 % of the problems result in fewer valid storm track connections compared to HA. For the mean error of NN, a peak is seen in a 250 km pruning radius. For larger radii and smaller problem sizes, geographical constraints cause the NN to have very few legal moves, leading to frequent detections of the optimal solution. As the problem size grows, the NN has more possible assignments, and thus more opportunities to deviate from the optimal, which is reflected in the increased number of suboptimal solutions. For

large problem sizes, few missed connections by NN will result in a relatively small error percentage. This explains why, as the size of the problem increases, the mean error percentage decreases since deviations from the optimum become increasingly possible, but their relative contribution to the total problem cost is smaller. In summary:

- For small problems, suboptimal NN solutions are rare, but each one has a large relative effect.
- For medium problems, suboptimal NN solutions are more frequent and have a significant impact, producing the peak in mean error.
  - For large problems, suboptimal NN solutions are very frequent, but have a smaller relative effect, lowering the mean error.

These patterns are further emphasised by the decrease in the ratio of missed connections versus suboptimal solutions, as the size of the problem increases, from  $29\% \left(\frac{7}{24}\right)$  for a  $250 \,\mathrm{km}$  radius over  $21\% \left(\frac{21}{99}\right)$  for  $175 \,\mathrm{km}$  to  $19\% \left(\frac{126}{658}\right)$  for a  $100 \,\mathrm{km}$  radius.

Although the HA is significantly slower at small radii (e.g. 9 s at 100 km), it consistently achieves more complete and accurate tracks, critical for avoiding premature track termination. In applications where the domain is large (e.g. global reanalyses), similar complexity issues could potentially arise even with larger radii, making global optimisation increasingly relevant.

#### 285 4.2 Effect of track reconciliation

290

295

To evaluate the effect of the BLOB analysis technique for track reconciliation, we compare the full tracking system with and without the reconciliation step (see Table 3). This comparison does not assess the physical accuracy of the reconciled tracks as such. Instead, it quantifies how often reconciliation is applied and how it influences the properties and survivability through post-processing filters for the final track dataset. The evaluation metrics are as follows:

- 1. The number of impact events in which at least one storm track survived post-processing out of the 37 total events.
- 2. The number of events with at least one surviving track that has used reconciliation out of the total number of events with at least one surviving track.
- 3. The number of tracks where reconciliation was used out of all surviving tracks in all events.
- 4. The number of reconciliations across all the surviving tracks versus the number of surviving tracks where reconciliation was used.
- 5. The average increase in track duration for tracks where reconciliation was used.
- 6. The average increase in track length for tracks where reconciliation was used.

Using the pruning radius of 350 km as an example, we notice that out of 37 total events, we have 36 events where at least one track survived the post-processing (see Table 3). From here we have found that the BLOB utilisation has played a role in

**Figure 7.** Locations where the track reconciliation step successfully connects fragmented storm tracks. The plot shows 21 surviving, reconstructed ETC tracks, with different colours representing individual cyclone systems. Identical colours indicate parts that were merged into a single continuous track. Numbers mark the time steps at which reconciliation happens for pruning radius 350 km.

**Table 3.** Summary of track reconciliation performance across different pruning radii. Metrics include the number of events with detected tracks, events and tracks where reconciliation is active, total number of reconciliations, and improvements in track duration and distance.

| Metric \ pruning radius [km]              | 700   | 500   | 350   | 250   | 175   | 100   |
|-------------------------------------------|-------|-------|-------|-------|-------|-------|
| 1. Events with a track                    | 36/37 | 36/37 | 36/37 | 37/37 | 36/37 | 34/37 |
| 2. Events where reconciliation is active  | 11/36 | 15/36 | 19/36 | 21/37 | 26/36 | 28/34 |
| 3. Tracks where reconciliation is active  | 13/42 | 15/41 | 21/43 | 24/47 | 28/43 | 31/38 |
| 4. Total number of reconciliations        | 14/13 | 16/15 | 30/21 | 43/24 | 51/28 | 70/31 |
| 5. Average increase in track duration (%) | 44    | 52    | 76    | 86    | 83    | 123   |
| 6. Average increase in track length (%)   | 33    | 45    | 59    | 64    | 72    | 96    |

19 of these events, where over the course of these 19 events, 21 surviving tracks have been affected. In the 21 post-processed tracks, 30 reconciliations have occurred, which means that multiple surviving tracks have had more than one reconciliation. In summary, the use of BLOB analysis has ensured that a significantly higher number of tracks have survived post-processing. All of the 19 events where BLOB analysis has been utilised have been manually investigated on an individual basis, and it has been found that the vast majority of the tracks in these events would otherwise not have met the criteria of post-processing.

Figure 7 shows the geographical locations where BLOB reconstructions have occurred for all 37 events. Most of these are located on complex terrain, mountains and land-ocean boundaries, which means that it works as intended. When a reconstruction is made, there can in some cases be a large distance between two points that are connected, and such large jumps will show up in the final storm track. It might be possible to define a smoothing framework that handles these jumps, but this is not trivial, and we will leave this for further developments in future versions of ImpactETC.

## 310 4.3 Sensitivity and optimization of post-processing parameters

In order to evaluate the sensitivity of the post-processing parameters and define a set of optimal parameters for our case study, a full grid search exploration of the parameter space is carried out and the post-processing results are recorded. The parameter ranges applied for the grid search are as follows:

- Minimum travel distance: [0 km, 100 km, ..., 1400 km, 1500 km]

- Minimum duration: [0 h, 1 h, ..., 39 h, 40 h]

-  $\Delta$  AoR:  $[-5^{\circ}, -4^{\circ}, ..., 9^{\circ}, 10^{\circ}]$ 


- Minimum time in AoR: [0 h, 1 h, ..., 29 h, 30 h]

We note that the AoR is centred on 60° lat, 15°N and initially spans from 50 – 70° lat and 0 – 30°E. Here, the AoR change is in degrees with negative values decreasing the AoR and vice versa. The initial AoR encompasses the Baltic region and the North Sea (see the area highlighted in Figure 1), since an ETC must enter this region to affect the water levels on the Danish coasts.






To explore the parameter sensitivity, we use two metrics. The first metric evaluates whether the post-processing filters result in the true number of ETC tracks per event for our case data. To do this, we have gone through the labour-intensive work of manually labelling the correct number of ETC tracks within the AoR for all impact events. We then define a simple accuracy score for the evaluation, which we call the "True Accuracy" (TA):

$$TA = \frac{1}{n} \sum_{i=1}^{n} \mathbf{1}(y_i = \hat{y}_i)$$
 (2)

where y is the true number of ETCs,  $\hat{y}$  the predicted number given a set of post-processing parameter values,  $\mathbf{1}(y_i=\hat{y}_i)$  is an indicator function equal to 1 if correct and 0 if incorrect, and n is the total number of events. Users of ImpactETC1.0 can do the same for their case implementations if the number of events to investigate is relatively small and manual labelling is deemed possible.

We recognise that developing a labelled dataset may not be desirable or possible for all applications of the algorithm, especially when evaluating many impact events. In order to be able to still calibrate post-processing parameters rather than simply choosing values on an ad-hoc basis, we test a second metric that does not require any manual work. If this second metric is able to calibrate parameter values similar to those of the True Accuracy, it can be used for large automatised analyses. We define this second metric as the "Single Storm Score" (S):

$$S = \frac{N_1 - N_0 - N_{3+}}{N_0 + N_1 + N_2 + N_{3+}} \tag{3}$$

where  $N_i$  represents the number of events observed with i associated ETC tracks. This score rewards events with exactly one detected ETC (interpreted as a successful and unambiguous match), penalises missed detections ( $N_0$ ) and excessive ambiguity ( $N_{3+}$ ). It treats cases with two ETCs as neutral, since we have observed that this can sometimes happen in our case example. The result is a simple, interpretable score ranging between -1 and 1, where higher values indicate better post-processing performance.

Figure 8 shows the results of the parameter sensitivity analysis with both metrics. Each subplot in the figure shows the sensitivity of the ETC detection to one of the four post-processing parameters, respectively. Given the discretization of the chosen grid search for each parameter, there are a total of 325,376 possible combinations of the four parameters. The plots show the median, the 5th and the 95th percentiles of the score values across all of these combinations. The results show that ETC detection is most sensitive to the size of the AoR (Figure 8a), time in AoR (Figure 8b), and the minimum travel distance (Figure 8c). Notably, there is the special case where minimum time in AoR equals 0, meaning that an ETC does not need to enter the AoR for detection, which leads to a large drop in accuracy (Figure 8b). The large distance between the 5th and 95th percentiles shown by the shaded areas indicates that the same parameter value can lead to very different post-processing performance depending on the values that the other post-processing parameters take. Thus, there is a large dependence between parameters, e.g. the size of the AoR and the minimum time in the AoR are closely related.



Figure 8. Sensitivity of the ImpactETC1.0 algorithm's performance to key post-processing parameters. Each panel shows the impact of varying a specific parameter on two metrics: the S score (black) and TA relative to historical impact records (blue). (a) Sensitivity to  $\Delta$  AoR in degrees, (b) number of time steps required within the AoR, (c) minimum travel distance of detected ETC tracks in km, and (d) minimum ETC duration in hours. These experiments highlight trade-offs in tuning filtering parameters.

Figure 9 shows the marginal and joint distributions of the True Accuracy and Single Storm Score after the values of both scores have been min-max normalized to fit the same 0-1 range. The marginal distributions follow each other closely with small deviations in some ranges. The joint distribution confirms that the Single Storm Score produces normalized values that are similar to the True Accuracy. This, combined with the general agreement of the scores in Figure 8, shows that the Single Storm Score is good at approximating the True Accuracy and thus suitable for impact-focused calibration when manual labelling is not possible.

For the presented case on ETC's that produce storm surge events, we select the parameter set that gives the best Single Storm Score. Given the grid search discretisation, there are 325,376 possible combinations of the four post-processing parameters. In the presented case, 20 unique parameter sets gave the same score. We selected the parameter set that was closest to the median

**Figure 9.** Comparison of marginal (left) and joint (right) distributions of the True Accuracy and the Single Storm Score after both scores have been min-max normalized to fit 0-1 ranges. The joint distribution is shown as corresponding True Accuracy values for discretized bins of Single Storm Score values. The red dashed line indicating the ideal 1:1 relationship. Colour shading in box plots indicates the number of observations per bin.

of each parameter (see values in Table 4), thus avoiding extremities or fortunate local minima by accident. We note that this method for selecting parameter combinations is only suitable if the possible combinations are centred in nature. If they had, e.g. been clustered in two different areas of the parameter space, the user would have to devise another way to select the final set. It is worth noting that even though all 325,376 parameter combinations were simulated in a brute-force manner, this still only took  $\tilde{3}$  minutes to complete on a standard laptop. This highlights that calibration of post-processing parameters is a simple and fast step that should be performed.

Table 4. Optimal post-processing parameter values identified for the storm surge case in this study.

| Minimum travel distance (km) | Minimum duration (h) | Minimum time in AoR (h) | $\Delta$ AoR (degrees) |  |
|------------------------------|----------------------|-------------------------|------------------------|--|
| 200                          | 31                   | 12                      | 0                      |  |

## 5 Discussion



#### 5.1 Strengths and limitations of the track algorithm

To check the quality of the identified ETC tracks, we here present a set of representative storm surge cases. Figure 10 shows six examples of well-defined ETC tracks that exhibit consistent spatial progression, relatively smooth trajectories, and alignment with the underlying MSLP field. These cases demonstrate the ability of the algorithm to capture the temporal and spatial evolution of ETCs, even as they traverse complex regions with steep topography and land-ocean boundaries. The 2002-01-28







event shows a track where BLOB-based track reconciliation was performed three times. Without the reconciliation step, there would have been four small fragmented tracks that would have been eliminated by the post-processing step.

In contrast, Figure 11 highlights six events that illustrate the challenges of the algorithm. The 2000-01-29 event has a track with discontinuities or unrealistic abrupt changes in direction, associated with the ETC moving over complex topography. The 2013-10-28 and 2013-12-05 events are correctly identified, but so are a couple of secondary ETC tracks that are impact-irrelevant, which then have to be manually removed. The 2015-11-29 identifies an impact-irrelevant track, while the actual relevant track (highlighted in red) is filtered out because its duration is slightly shorter than the calibrated post-processing parameter for minimum ETC duration. This shows that a single set of parameters in the post-processing step cannot universally discriminate the right amount of ETC's but has to balance the filtering of short-lived yet impact-relevant storms, while minimising the amount of secondary irrelevant tracks. An interesting situation happens in the 1993-12-19 event where the MSLP contains a large ETC identified in blue with two secondary ETC's that are short-lived and thus filtered out. However, one of the secondary ETC's is actually impact-relevant since it produces the steep pressure gradients over Denmark that cause the impact. Finally, the 2016-12-26 event shows a case where the track reconciliation step fails. As the ETC (blue track) approaches the Norwegian coastline, large MSLP artefacts appear in the reanalysis dataset, which leaves two potential track pathways. The BLOB analysis makes the wrong choice and connects to the Northern option, instead of the series of fragmented segments in red that continue south-east and cause the impact.

These visual assessments confirm that the proposed framework effectively captures and isolates the ETCs driving extreme impacts. It also, as expected, reveals situations where complex terrain, MSLP artefacts, or track fragmentation can degrade performance and present opportunities for future research to handle.

## 5.2 The value of global optimisation to solve the correspondence problem

A central question when choosing between a heuristic or an exact solution for the correspondence problem is whether the additional computational cost of global optimisation meaningfully impacts overall algorithm performance. Table 5 shows the total computational run times for using ImpactETC1.0 on all 37 events serially. The run time is divided by the main steps of the framework and tested for several different pruning radii, i.e. r-values, which leads to different sizes of problem complexities to solve. This analysis shows that solving the correspondence problem with the HA accounts for only a small fraction of the total computational time, especially compared to more time-consuming components such as data loading and candidate point identification. Even at the highest level of correspondence complexity (100 km pruning radius), solving the correspondence problem remains a minor contributor to total run time compared to candidate point processing and BLOB analysis. This suggests that the modest additional cost of the Hungarian Algorithm does not justify compromising track accuracy by using a nearest-neighbour heuristic. Since the analysis in Table 2 shows that the NN heuristic errors drastically increase with the complexity of the problem, using NN-based heuristics to solve the correspondence problem could become increasingly problematic as the resolution of reanalysis datasets improve in the future. High-resolution reanalyses can resolve small secondary ETC and several local minima in vague MSLP fields, which increases the risk of making wrong track connections. This is something

**Figure 10.** Six examples of well-defined, successfully identified ETC tracks. Blue dots mark the time evolution along each track, and stars indicate the initial position of each track. The background shows the MSLP field at the median time step of each track, providing synoptic context for storm positioning and structure. For each example, the location of the water level gauges that observe a storm surge are highlighted in red.

**Figure 11.** Six examples of ETC tracks that show the limitations of the algorithm, e.g. track splitting, short-lived centers, non-smooth movements, or secondary impact-irrelevant tracks that survive post-processing. Blue tracks are successfully identified ETCs, orange and green denote secondary impact-irrelevant tracks that survive post-processing, while red shows tracks that are impact-relevant but were filtered out. For each example, the location of the water level gauges that observe a storm surge are highlighted in red.


future research should investigate, and if it turns out to be a significant problem, we have shown here that global optimisation with the HA is a fast and accurate alternative.

**Table 5.** Breakdown of algorithm run time (s) across individual framework components for different candidate point pruning radii, r. Columns show total run time vs times spent on data loading, candidate identification, solving the correspondence problem, BLOB construction, and post-processing. Values are averages over all events, highlighting how pruning radius affects computational demands and the relative contributions of each algorithm component.

| Pruning     | Total run time [s]  | Data load [s] | Identify       | Solve              | BLOB             | Post-processing [s] |
|-------------|---------------------|---------------|----------------|--------------------|------------------|---------------------|
| radius [km] | Total full time [8] | Data toau [8] | candidates [s] | correspondence [s] | construction [s] | rost-processing [s] |
| 700         | 546.89              | 368.78        | 48.01          | 2.1                | 127.94           | 0.0                 |
| 500         | 565.8               | 357.53        | 57.7           | 1.93               | 148.56           | 0.01                |
| 350         | 669.01              | 361.05        | 90.09          | 2.29               | 215.52           | 0.01                |
| 250         | 806.75              | 378.8         | 141.07         | 2.78               | 284.04           | 0.01                |
| 175         | 964.37              | 358.87        | 222.5          | 3.8                | 379.14           | 0.01                |
| 100         | 1386.1              | 311.38        | 466.69         | 12.22              | 595.72           | 0.04                |

#### 5.3 Future improvements and considerations for broader application

While the ImpactETC1.0 framework demonstrates robust performance in identifying impact-relevant ETC tracks, there are several avenues to further improve its accuracy and computational efficiency.

Computational efficiency: Data loading consistently dominates runtime across all pruning radii, and while optimised data handling could reduce this component, it will likely remain a significant fraction of total runtime. The candidate identification step is already quite scalable, but it might be possible to avoid excessive block comparisons with further developments. For track reconciliation, there are likely gains to be made, e.g. by adding a distance-based pre-check before BLOB initialisation, which would reduce unnecessary BLOB constructions.

Improved tracking: In terms of the quality of the final tracks, more research could be done on how to refine the track reconciliation step, for example, as seen in Figure 11 for the 2016-12-06 event, where the BLOB analysis makes the wrong connection because there is a single time step where no candidate point is found on the correct track. It might be worth exploring how to incorporate multiple adjacent time steps in the BLOB analysis instead of limiting reconciliation checks to simultaneous candidate points. It could also be worth testing the implementation of user-defined masks of high-uncertainty areas such as mountain ranges, where the BLOB analysis is given more freedom in space and time to make the right connections. This extension would enable a more realistic identification of track jumps across complex terrains or during rapid ETC evolution. There may also be improvements by using ETC tracking information at higher pressure levels, such as 700 hPa, to improve the continuity of surface-level tracks during reconciliation. While the BLOB-based reconciliation step is crucial to avoid premature track termination, it sometimes leaves irregular jumps between points on the track. It would be interesting to explore ways to smoothen tracks around reconciliation jumps. This could potentially also be guided by information at higher pressure levels.






Post-processing: The example events in Figure 11 showed areas of improvement for the post-processing step. In the cases where multiple tracks are found within the AoR, there could be further steps that estimate which of the tracks are actually impact-relevant. An option might be to select the track that is geographically closest to the impact location at the time of impact. However, it would also be necessary to devise a check on whether an impact might actually be caused by the interaction of multiple ETC's within the AoR. Figure 11 also showed one event in which the impact-relevant ETC was filtered out because its total duration was too short. Hence, a potential addition to the algorithm could be checking for filtered impact-relevant tracks. Additional features: To deepen the physical interpretation of ETC evolution, associating additional properties - such as MSLP at ETC centre, vorticity, wind speed, or gusts — along identified tracks would allow classification of different stages of

Broader applications: We believe that ImpactETC1.0 is broadly applicable to other case areas. Users applying the framework to other reanalysis datasets or to different geographic regions should consider recalibrating key parameters, especially candidate detection thresholds, correspondence distance limits, and AoR size, to account for differences in data resolution, domain size, and regional storm climatology.

the storm life cycle, including genesis, intensification, mature phase, and occlusion.

#### 440 6 Conclusion

This study introduces ImpactETC1.0, a novel framework for tracking extratropical cyclones (ETCs) with a specific emphasis on linking storm tracks to observed surface impacts. The framework incorporates several innovative components, including global optimisation via the Hungarian Algorithm, BLOB-based track reconciliation, and calibrated post-processing procedures. These methodological advances enable an accurate reconstruction of the ETC tracks, even in regions with complex topography and during phases of rapid storm development, while maintaining relatively low computational demands.

Using a variant of a nearest-neighbour heuristic when solving the correspondence problem is standard practice for ETC tracking algorithms. Such heuristics are fast, but come with the risk of making wrong connections. In this study, a comparison between the Hungarian Algorithm and the nearest-neighbour heuristic highlights that in scenarios with many candidate points that have to be connected over time, the nearest-neighbour heuristic creates suboptimal solutions in 33 % of the cases. The Hungarian Algorithm's global optimisation requires an order of magnitude more computation time, but is still very fast and accounts for less than 1 % of the total algorithm's run time, which is dominated by candidate point selection and track reconciliation. Our results suggest that this issue could become increasingly significant for the general scientific field of ETC tracking in increasingly higher resolution reanalysis datasets, and that future research should investigate this question further.

As it can be difficult to perform surface-level tracking of ETC's over complex terrain, this study developed and tested a BLOB analysis-based step for reconciling fragmented tracks. The results show that such a reconciliation step is necessary when ETC's traverse complex terrain such as the Scandinavian mountains and land-ocean boundaries. The issue of fragmented tracks grows when there are many candidate points that need to be connected over time, which suggests that this issue will also increase in importance for high-resolution reanalysis datasets. In the tested case, reconciling fragmented tracks results in final ETC tracks that are on average twice as long, as it prevents premature termination. The BLOB analysis used in this



460 study represents a novel contribution to the storm tracking literature, indicating the potential for further improvements and applications in future research.

Additionally, an impact-oriented calibration of post-processing parameters lead to the removal of unnecessary tracks while ensuring an increased correspondence between identified ETC tracks and the impact events, which enhances the applications for impact studies. The study investigated two metrics to calibrate the parameters. One of these was based on labour-intensive manual labelling of the correct number of ETCs per impact event. The other used automatic scoring in the form of a metric named the Single Storm Score, which penalises both under- and overestimation of the resulting number of ETC tracks. The results show that the Single Storm Score produces calibrated parameter values similar to those obtained by manual labelling. These results demonstrate that it is possible to calibrate post-processing parameters for impact-focused tracking without performing a large amount of manual event inspection.

Proposed future improvements include using vertical storm information to improve the tracking and reconciliation of fragments, applying additional post-processing steps to improve the discrimination of tracks related and unrelated to the impact, and enhancing the speed and efficiency of key algorithmic steps.

ImpactETC1.0 has the potential to be used more universally for storm tracking applications in other geographical regions, provided it is calibrated to those conditions. This offers a powerful tool for advancing, e.g. climate hazard assessments and operational forecasting capabilities.

## 7 Appendix A

Code and data availability. The codes and the dataset used in this paper are available at Zenodo: https://doi.org/10.5281/zenodo.17093393 (Agertoft et al., 2025). The CERRA sub-daily regional reanalysis data used in this study are freely available from the Copernicus Climate Change Service Climate Data Store (C3S CDS): https://doi.org/10.24381/cds.622a565a (Schimanke et al., 2021).

Author contributions. NA conceptualized the tracking framework, developed the methodology and software, performed the analysis, made visualizations and wrote the initial draft. JS handled data acquisition, data analysis, and visualizations. JP conceptualized the framework, data analysis and interpretation. IR performed data analysis. ML acquired funding and was the project coordinator for study conception and design. All authors revised and edited the manuscript. All authors have read and approved the manuscript.

Competing interests. The contact author has declared that none of the authors has any competing interests.

| start_date          | end_date            |
|---------------------|---------------------|
| 1991-01-09 07:00:00 | 1991-01-09 12:15:00 |
| 1991-12-20 11:00:00 | 1991-12-20 19:00:00 |
| 1992-01-17 09:00:00 | 1992-01-17 09:00:00 |
| 1993-01-12 23:15:00 | 1993-01-12 23:15:00 |
| 1993-02-21 16:00:00 | 1993-02-21 22:45:00 |
| 1993-12-19 23:30:00 | 1993-12-20 00:00:00 |
| 1996-11-06 07:30:00 | 1996-11-06 07:30:00 |
| 1997-10-11 18:45:00 | 1997-10-11 18:45:00 |
| 1998-10-25 22:00:00 | 1998-10-25 22:00:00 |
| 1999-12-03 15:45:00 | 1999-12-03 18:30:00 |
| 2000-01-29 19:45:00 | 2000-01-30 04:00:00 |
| 2000-10-30 17:00:00 | 2000-10-30 23:00:00 |
| 2002-01-28 22:00:00 | 2002-01-28 23:10:00 |
| 2002-02-21 00:10:00 | 2002-02-21 10:50:00 |
| 2002-02-26 18:50:00 | 2002-02-26 19:40:00 |
| 2003-12-06 15:16:00 | 2003-12-06 21:30:00 |
| 2004-11-18 16:10:00 | 2004-11-18 16:10:00 |
| 2005-01-08 14:40:00 | 2005-01-09 01:40:00 |
| 2006-11-01 16:40:00 | 2006-11-02 09:50:00 |
| 2007-01-12 08:00:00 | 2007-01-12 08:00:00 |
| 2007-01-19 06:40:00 | 2007-01-19 06:40:00 |
| 2007-01-25 02:00:00 | 2007-01-25 02:00:00 |
| 2008-03-01 20:00:00 | 2008-03-02 03:30:00 |
| 2011-02-11 22:10:00 | 2011-02-11 22:10:00 |
| 2011-11-27 18:20:00 | 2011-11-27 20:50:00 |
| 2011-12-09 18:40:00 | 2011-12-10 04:40:00 |
| 2012-01-14 06:10:00 | 2012-01-14 08:20:00 |
| 2013-10-28 15:30:00 | 2013-10-28 16:00:00 |
| 2013-12-02 03:20:00 | 2013-12-02 03:20:00 |
| 2013-12-05 15:00:00 | 2013-12-06 23:50:00 |
| 2015-01-10 11:40:00 | 2015-01-11 10:20:00 |
| 2015-11-29 18:50:00 | 2015-11-29 18:50:00 |
| 2016-12-26 19:10:00 | 2016-12-27 04:40:00 |
| 2017-01-04 16:50:00 | 2017-01-05 01:00:00 |
| 2017-10-29 20:00:00 | 2017-10-29 23:50:00 |
| 2019-01-02 04:30:00 | 2019-01-02 17:40:00 |
| 2019-01-09 02:10:00 | 2019-01-09 07:10:00 |

**Table A1.** Start and end dates of events, if only one station was hit then start date equals end date, if not then multiple stations were hit sequentially and start date represents the peak of the earliest station hit, and end date represents the peak of the last station hit in that event. Timestamp highlights peaks of event.

Acknowledgements. Niels Agertoft worked for the ECCO project, which received funding from Innovation Fund Denmark and the European Union's Horizon Europe Programme under the 2022 Joint Transnational Call of the European Partnership Water4All (Grant agreement ID: 101060874). Jonas Wied Pedersen was financially supported by the Innovation Fund Denmark through the industrial postdoc programme [grant number: 0197-00005B]. The authors thank ECMWF and Copernicus for making the CERRA dataset available.

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
