# Peer review of "ImpactETC1.0: Impact-oriented tracking of extratropical cyclones with global optimisation and track reconciliation"

_EGUsphere, 2025_

## Referee Comment (RC1)

The manuscript titled "ImacptETC1.0: Impact-orientated tracking of extratropical cyclones with global optimization and track reconciliation" introduces a new method of tracking extratropical cyclones in the Nordic region and linking them to storm surge impacts observed at water level recording stations in the region. The authors present a three step cyclone tracking process including an optimized global solution to connect storm centers in time, a BLOB method to connect storm tracks across mountainous terrain and land-sea barriers, and some post processing steps to identify cyclones of interest.

Overall, I think the manuscript presents some interesting ideas to the challenge of extratropical storm tracking, the most influential potentially being the so called 'BLOB method' used to track storms through discontinuous jumps across mountainous terrain. I think the manuscript could be greatly improved by providing a more detailed presentation of the results and reworking of certain thresholds defined in the method. I also think the results are lacking focus on the impacts of individual cyclones. Details of the magnitude of these events could be shown for example, and linking to specific storm or storms done through a wind and proximity analysis seems readily available. Therefore, I am recommending major revisions to the current work. Please find my detailed comments below.

Line: 83: "impact-irrelevant" is dependent on the specific aim of each study. Dynamically strong systems out at sea could be very hazardous to those traveling by ship or coastlines vulnerable to large swell. I'd argue few strong storms are without any impact.

Lines 82-85: I believe the method presented in this paper has many of the same issues described in these lines. The authors apply similar "simple" thresholds with respect to minimum duration, maximum MSLP center, and minimum track length, with a small addition of proximity to the impacted region. I would suggest softening this argument in the introduction.

Figure 1: An additional marker or box showing the domain of panel on the left in right panel map would be helpful to the audience

Line 136: Expanding this time window could help limit the amount of storms filtered out for short duration time later.

Figure 2: Do all sea level events have photography? Why de-tided? Tides have large impact on flooding whether high or low, neap or spring. A weaker extratropical cyclone could have large impacts if it occurs during a spring high tide.

Line 145-147: How is the upscaling done? Are you left with a 350km grid afterward? If not, how much coarser is the grid compared to the original 5.5km? Is the upscaled grid directly related to the pruning radius? In essence this is down sampling your grid, which makes sense given the spatial resolution is far smaller than typical scales of ETCs. Where is the location of the final minima placed within the grid?

Line 151: What is MSLPmax based on? Having a higher or lower background field of MSLP could affect the precision of this parameter. Could consider using the gradient or Laplace of MSLP to avoid this problem or as another check in your method.

Line 153: How do you know your relative vorticity intensity isn't driven by a strong frontal feature and possibly not associated with an ETC? The addition of a wind speed threshold in the same radius may be interesting as well. Also see Gramcianinov (2020) for tracking using relative vorticity

Gramcianinov, C. B., Campos, R. M., de Camargo, R., Hodges, K. I., Guedes Soares, C., & da Silva Dias, P. L. (2020). Analysis of Atlantic extratropical storm tracks characteristics in 41 years of ERA5 and CFSR/CFSv2 databases. Ocean Engineering, 216, 108111.

Figure 4: With this MSLP threshold you may miss storms if they're located in a higher background pressure field.

Please add lat/lon ticks on all axes.

Around the pressure minima shown in figure 4d, do you see closed isobars at 1mb intervals or do you see closed circulation in the wind field? I believe more validation of these cyclone minima is required.

Line 180: This implies your threshold for ETC speed is 600km/hr which is unrealistic compared to previous literature. How do you reconcile this or validate that these storm centers should be connected in time?

Lines 199-200: The Hungarian Algorithm seems to only be given information at 2 time steps (t and t+1), then connects the local minima between these time steps according to the shortest distance summed across all connections. What are the implications of this decision affecting downstream connects in time? This framework could lead to the premature termination of tracks by limiting future options for connection. In other words, if you chose a point further away from your minima at time t, you have different minima to choose from within your 600km distance threshold at time t+1, t+2, t+3 and so on. Would be great to see an animation of tracks being formed by the method and how the NN functions in comparison over the hourly evolving MSLP and wind field.

Line 218: Connecting candidate points that are 2500 km apart from on another seems excessive. The typical radius of an ETC is on the order of 500km. How sensitive is this process if you set the maximum distance between candidate points to 500km or 1000km? I'm concerned this will very likely lead to spurious connections across storms which should not be connected in time.

Line 235-240: I think a criteria related to the local winds and water level would improve this framework. Why not include a criteria of the storm center needing to be within a certain distance (maybe 500-1000km) of the impacted area during the peak of the surge/wind event?

Table 1. Why not define AoR using a radius of a certain distance around the impact location? There is also a question of the wind direction the impact location is experiencing. Depending on where you are in relation to the storm center being tracked you could be experiencing winds in opposite directions, which would directly impact the surge experienced. Adding such a criterion could help limit mis assigned storms.

Figure 7. Why are there storm centers that are within one 350km pruning radius of one another? Shouldn't these have been pruned or connected in time already given the 600km threshold of HA connections in time. Numbers 13,60,27, 73, 35.. all seem like their connecting 2 points in very close proximity to one another.

Table 2. I'm not convinced the optimal solution from HA is physically correct with regards to real world tracking, therefore I find it inappropriate to use it as ground truth to the nearest neighbor approach. How would this method compare to a small subset of hand drawn tracks?

283: This may be true in terms of minimizing distance summed across all tracks, but I don't see clear evidence of this method producing "More accurate tracks." This implies the shortest track is always the most accurate which seems like it wouldn't be true in all cases. One specific example is the case of a storm center splitting into 2, this tracking method would miss a potential storm track if one center was slightly further away at one time step, but was longer lasting overall. Wouldn't it be better to track both or perhaps even the longer lasting segment that was initially further away?

Line 298: What was the nature of the one event not tied to a storm track?

Line 315: Varying AoR by degrees lacks physical meaning as distance between lines of longitude will vary substantially depending on where you are in the domain. Why not use a kilometers-based approach?

Line 371: Alignment with the underlying MSLP field is very difficult to observe in the still images covering a wide time/space range. I think an animation in the supplementary materials would be more affective.

Line 375-376: These large jumps could be due to the large search radius of 600km, as this occurs while the storm center is still offshore but seems to be getting distracted by noisier MSLP minima along the coast, or perhaps the merging of another low-pressure system. Would choosing a smaller initial search radius and allow the BLOB method to connect the tracks after improve the result?

Figure 11 (Lines 377-378): how were orange and green tracks determined to be irrelevant? I think different post processing criteria could be applied to improve performance of the algorithm. Why not add a step that chooses the storm track closest to the impact location at the height of the event or something similar?

Line 384: how is the relationship between steep gradient and water level quantified? Is timing of the ETC and water level considered?

Line 430: I agree that more work should be done to assess the interaction of multiple ETCs and I think some of the conclusions drawn about storms being "impact irrelevant" may be premature here. Since the manuscript is focused on impact on sea level, I think some of these steps should be addressed here. It should be relatively low effort to apply one or two post processing steps to the current work.

431-432: Wind speed could also be investigated to limit the filtering out of impactful storms.

449: As previously stated, I think suboptimal is poorly defined and the global optimal solution should not be used as benchmark in this manner.

452-453: ETCs exist on timescales more than adequately resolved by the model being used here. One could even argue the high resolution is already unnecessary for resolving these large atmospheric features and could result in more confusion and runtime for algorithms as the number of small pressure perturbations increase. As resolution increases it is likely that model output would be down sampled spatially, much in the same way that the upscale gridding is performed here. I don't see the practicality of ETC tracking with higher resolution.

---

## Author Comment (AC2)

**Reviewer 1**

The manuscript titled "ImacptETC1.0: Impact-orientated tracking of extratropical cyclones with global optimization and track reconciliation" introduces a new method of tracking extratropical cyclones in the Nordic region and linking them to storm surge impacts observed at water level recording stations in the region. The authors present a three step cyclone tracking process including an optimized global solution to connect storm centers in time, a BLOB method to connect storm tracks across mountainous terrain and land-sea barriers, and some post processing steps to identify cyclones of interest.

Overall, I think the manuscript presents some interesting ideas to the challenge of extratropical storm tracking, the most influential potentially being the so called 'BLOB method' used to track storms through discontinuous jumps across mountainous terrain. I think the manuscript could be greatly improved by providing a more detailed presentation of the results and reworking of certain thresholds defined in the method. I also think the results are lacking focus on the impacts of individual cyclones. Details of the magnitude of these events could be shown for example, and linking to specific storm or storms done through a wind and proximity analysis seems readily available. Therefore, I am recommending major revisions to the current work. Please find my detailed comments below.

We appreciate the reviewer's careful evaluation of both the methodological framework and the presentation of the results. In particular, we value the reviewer's comments regarding (i) the need for a clearer and more detailed presentation of the tracking results, (ii) the definition and justification of several key thresholds used in the algorithm (including their sensitivity and physical interpretation), and (iii) strengthening the connection between tracked cyclone events and the observed storm-surge impacts at tide gauge stations. We will revise the manuscript accordingly, including clarifications and additional discussion of parameter choices, improvements to figures and post-processing descriptions, and additional analyses/illustrations that better highlight the magnitude and characteristics of impactful cyclone events. Below, we respond to each comment point by point and describe the changes we will make in the revised manuscript.

**R1C1** Line: 83: "impact-irrelevant" is dependent on the specific aim of each study. Dynamically strong systems out at sea could be very hazardous to those traveling by ship or coastlines vulnerable to large swell. I'd argue few strong storms are without any impact.

We agree with the reviewer that "impact-relevance" is dependent on the scope of the study. Our use of the term was intended to distinguish between cyclones that trigger a specific higher-end stakeholder-relevant event (in this case, coastal storm surges) and those that do not.
We chose storm surges (i.e., sea levels) as our primary indicator of "impact"

because they provide a physically consistent reference point to initialise the tracking algorithm. However, this framework is not limited to surges; if the focus were on inland wind damage or shipping hazards, the algorithm would function similarly provided a "time of impact" is specified. We will clarify in the Introduction and the Discussion that "impact" in this paper is exemplified by surges, but the framework is designed to be broadly applicable to other hazard types.

Notice also that we, as a result of this review stage, revise the algorithm and corresponding event list input data, to also include the location of impact so that the nearest storm track relevant for each event is detected.

**R1C2** Lines 82-85: I believe the method presented in this paper has many of the same issues described in these lines. The authors apply similar "simple" thresholds with respect to minimum duration, maximum MSLP center, and minimum track length, with a small addition of proximity to the impacted region. I would suggest softening this argument in the introduction.

We will soften this specific argument accordingly.

**R1C3** Figure 1: An additional marker or box showing the domain of panel on the left in right panel map would be helpful to the audience

Good suggestion. We will add the domain of the left panel to the right side panel in Figure 1.

**R1C4** Line 136: Expanding this time window could help limit the amount of storms filtered out for short duration time later.

We agree with the reviewer to the extent that expanding the time window could serve to make recognition of some tracks easier during post-processing, as it has for some cases been observed that the correct track begins at the first investigated time step. Increasing the time window could potentially highlight these correct tracks further. However, some tracks are not long-lasting in nature, and therefore expanding the time window could lead to additional, locally impact-irrelevant tracks appearing with the characteristics of a correct track, at least in terms of the time span of the track. Thus, expanding the time window around the timing of the impact-peak has the potential downstream ramification of filtering out correct tracks from other events during post processing. In the revised manuscript we will highlight these trade-offs.

**R1C5** Figure 2: Do all sea level events have photography? Why de-tided? Tides have large impact on flooding whether high or low, neap or spring. A weaker extratropical cyclone could have large impacts if it occurs during a spring high tide.

Fundamentally, the ETC tracking algorithm can be used on any type of event,

not just storm surge events, so the choice of de-tiding is not so important for this paper. Here, we simply want to use the case of storm surge events in Denmark to present an application of the tracking algorithm. We have chosen to de-tide the water levels to ensure we examine significant surge events caused by severe storms rather than tidal influence, which varies significantly across Danish Coastlines. Therefore, the timing of storm surges, in e.g. the Wadden sea, in relation to the tidal phase, is very crucial for the severity of the peak. By de-tiding, this influence was removed from the algorithm's development. If one would want to study the conditions of storms that produce surges along Danish coasts (which we actually aim to do in future research), then a discussion of tides vs wind-induced surges is very relevant. The photography in the figure is simply a visualization of an impact that has happened - it is only meant for this algorithm overview figure and not for any scientific use in this paper.

**R1C6** Line 145-147: How is the upscaling done? Are you left with a 350km grid afterward? If not, how much coarser is the grid compared to the original 5.5km? Is the upscaled grid directly related to the pruning radius? In essence this is down sampling your grid, which makes sense given the spatial resolution is far smaller than typical scales of ETCs. Where is the location of the final minima placed within the grid?

We acknowledge that our own choice of the word "upscaling" in the manuscript text can mislead the reader here. We do not "upscale" the grid in the sense that we coarsen the grid resolution. Instead, we start by splitting the full CERRA domain into non-overlapping square boxes with diagonal lengths of 350 km. We then perform our minima localisation on the original 5.5 km grid cells inside each box, which results in a single minimum within each box. Afterwards, we assess whether the minimum in each box is within a certain distance (the pruning radius parameter) of identified minima in nearby boxes. If the location of a point is within the pruning radius of another minimum, then the weaker of the two minima is eliminated as a potential candidate point. We save the latitude and longitude coordinates for the minima as well, so that the location of the final minimum in each spatial square is the 5.5 km grid cell that has the lowest value, thus remaining true to the original spatial resolution without upsampling. We'll revise the description to reflect this.

**R1C7** Line 151: What is MSLPmax based on? Having a higher or lower background field of MSLP could affect the precision of this parameter. Could consider using the gradient or Laplace of MSLP to avoid this problem or as another check in your method.

It is correct that the background field will affect the precision of this parameter. However, our choice of a relatively simple maximum cap on MSLP values (MSLPmax) is to simply filter away a lot of non-relevant local minima that stem from either noise in the data or very weak minima that we are confident cannot produce significant impacts. The reviewer is right that it would be possible to implement a stronger filtering step through additional checks e.g. based on the gradient or Laplace of MSLP. Our choice of a simple maximum cap is motivated by the fact that this is extremely computationally efficient, which we deem important as the "Identify candidate points" step of the algorithm already is one of the computationally demanding steps in the algorithm. In the case of storm surges in this paper, we have intentionally set the value of MSLPmax at a conservative value of 1010 hPa. We will revise the manuscript to highlight this.

**R1C8** Line 153: How do you know your relative vorticity intensity isn't driven by a strong frontal feature and possibly not associated with an ETC? The addition of a wind speed threshold in the same radius may be interesting as well. Also see Gramcianinov (2020) for tracking using relative vorticity. Gramcianinov, C. B., Campos, R. M., de Camargo, R., Hodges, K. I., Guedes Soares, C., & da Silva Dias, P. L. (2020). Analysis of Atlantic extratropical storm tracks characteristics in 41 years of ERA5 and CFSR/CFSv2 databases. Ocean Engineering, 216, 108111.

Thank you for the reference to Gramcianinov et al. We do not know if the relative vorticity is from a strong frontal feature, which might be true for a given case. As with the previous comment on the MSLPmax parameter, it would be possible to implement a stronger filtering step here than a simple vorticity threshold. Our choice is again motivated by computational speed. A wind speed threshold could potentially be interesting, but we believe it might be complicated to define a threshold value that is useful over both ocean and land. We are worried that it may filter away candidate points over land that we would consider interesting. We'll revise the manuscript to highlight this.

**R1C9** Figure 4: With this MSLP threshold you may miss storms if they're located in a higher background pressure field.

We believe this comment is similar to R1C7, see the full response up there. In short, we have chosen a quite conservative MSLP threshold value of 1010 hPa, which does not filter away impactful ETC's, and we will explain this in the revised manuscript.

**R1C10** Please add lat/lon ticks on all axes.

Good suggestion. This will be added to all figures.

**R1C11** Around the pressure minima shown in figure 4d, do you see closed isobars at 1mb intervals or do you see closed circulation in the wind field? I believe more validation of these cyclone minima is required.

When examining the need for validation of cyclone minima, we see two trade-offs that are relevant to discuss. One is about computational speed of the algorithm (the number of validation steps vs. the computations needed to perform these), and the other is about how conservative one wishes to be in terms of filtering potential candidate points (essentially the ratio between type 1 and type 2 errors). Based on several previous reviewer comments as well as this one, it seems that the reviewer generally prefers stronger filtering of minima than we have chosen to implement in this algorithm. The argument being that there if we let too many minima "survive" this step, then it will be more difficult and potentially introduce errors when we connect points through time during tracking. We believe this to be a valid concern. However, we would like to refer to our analysis of computational requirements of each algorithmic step (presented in the original manuscripts Table 5). Here, "data load" and "identify candidates" already consume significant proportions of the total algorithmic run time. Additional validation checks, especially those that require computation of derived fields (e.g. Laplacian or gradients) or loading of additional reanalysis fields (such as wind speed), would significantly increase the total algorithmic run time. We therefore prefer to err on the side of making type 2 errors, i.e. allowing more minima to pass through identification step. We are confident that the two super fast steps of the algorithm (tracking through time and post-processing) instead will handle and filter out these potentially non-relevant minima at a fraction of the computations it would require to handle them through more thorough validation early on. This discussion is highly relevant to the manuscript, and we will make sure that is addressed more thoroughly in the revised manuscript. To do so, we also plan to show how the issue of closed isobars and wind field circulation with the figure below for the specific event in Figure 4d. At this level (1 mb), we see closed isobars, closed circulation in the wind field, and significant wind speed levels for about half of the identified minima at this specific time step.

[Figure]

**(d) Vorticity-Filtered Minima**
(High Vorticity Regions) - Wind Speed & Direction with MSLP

**R1C12** Line 180: This implies your threshold for ETC speed is 600km/hr which is unrealistic compared to previous literature. How do you reconcile this or validate that these storm centers should be connected in time?

We agree that 600 km/h is not a physically realistic speed for the movement of an ETC. However, in the context of high-resolution reanalysis data (such as CERRA) and complex terrain, the $D_{max}$ parameter functions more as a "search radius" for a diagnostic variable (MSLP minimum) than as a physical speed limit. In regions with complex topography, the identified MSLP minimum can "jump" between grid cells, relative to the location of the synoptic system, from one hour to the next due to local pressure perturbations or artefacts, even if the synoptic system itself is moving at a standard speed. While we also have a subsequent BLOB-based reconciliation step to fix these issues, that step is much more computationally demanding than both HA and NN, and we therefore aim to handle minor jump issues in this step here, as this is significantly more efficient. We have experimented more with the parameter values for $D_{max}$ and checked the effects on the final tracks. Based on this, we will, in fact, recommend a lower value in the range 200-300 km in the revised manuscript, where we will also improve our discussion on how to set this parameter for a given use case. 200-300 km is still a bit larger than what other tracking algorithm case studies often employ (which seems to be in the range of 100-200 km per hour), but such a difference is related to the aforementioned problems with MSLP over complex terrain.

**R1C13** Lines 199-200: The Hungarian Algorithm seems to only be given information at 2 time steps (t and t+1), then connects the local minima between these time steps according to the shortest distance summed across all connections. What are the implications of this decision affecting downstream connects in time? This framework could lead to the premature termination of tracks by limiting future options for connection. In other words, if you chose a point further away from your minima at time t, you have different minima to choose from within your 600km distance threshold at time t+1, t+2, t+3 and so on. Would be great to see an animation of tracks being formed by the method and how the NN functions in comparison over the hourly evolving MSLP and wind field.

It is correct that the HA implementation (as well as the NN benchmark) only receives information from two time steps at a time. If this was the final step for connecting points through time and thus building the tracks, there could be an issue with tracks breaking and premature termination, as argued by the reviewer. However, the BLOB step that follows the HA optimization, handles this issue if the user chooses a parameter value for "Max BLOB candidate point distance" that is larger than $D_{max}$, which is what we recommend and use ourselves. In this case, the BLOB step will realise that a track has prematurely ended and join it to the remainder of the track. We see that our motivation for the BLOB step is mainly focused on complex terrain issues in the current text. We will make sure that the revised manuscript also explains that there is a potential issue with premature termination from only connecting points across two sequential time steps, but that the BLOB step will handle this issue if used appropriately. The request for animations is good and will improve understanding of the algorithm. We will provide these as well for the revised manuscript.

**R1C14** Line 218: Connecting candidate points that are 2500 km apart from on another seems excessive. The typical radius of an ETC is on the order of 500km. How sensitive is this process if you set the maximum distance between candidate points to 500km or 1000km? I'm concerned this will very likely lead to spurious connections across storms which should not be connected in time.

This comment is related to the discussion in the comment before this one (R1C13). The "Max BLOB candidate point distance" parameter that the reviewer here refers to, should be high enough for the algorithm to be able to fix track splitting and premature termination caused by, e.g. complex terrain, yet not be so large that spurious connections can be made. Upon further consideration, we agree with the reviewer that the choice of 2500 km is somewhat excessive. In tested set of events that we present in the paper, the largest connection that is made by the BLOB step is $\tilde{6}00$ km, and therefore the parameter can be lowered to this value without any changes to the results. For the revised manuscript, we will update our discussion and recommendations for this parameter, including a description of how sensitive the resulting tracks are

to this parameter.

**R1C15** Line 235-240: I think a criteria related to the local winds and water level would improve this framework. Why not include a criteria of the storm center needing to be within a certain distance (maybe 500-1000km) of the impacted area during the peak of the surge/wind event? Table 1. Why not define AoR using a radius of a certain distance around the impact location?

There are three separate suggestions in this comments (1: local winds/water levels, 2: max distance to between ETC center and impact location, 3: define AoR as a circle around impact location):

1. In our framework, we start from a known impact (here a storm surge) for which the user provides the location and time. We therefore do not need an additional step on local water levels (or winds) to identify if an impactful event has happened and if there is a relevant ETC that needs to be tracked for the event. The user has already made that decision.

2. We believe that the use of an AoR and associated requirements to how many time steps an ETC spends within the AoR is very related to including a criterion that the ETC centre should be within a certain distance at the time of impact. Rather than being within a certain distance (here 500-1000km), the ETC center needs to be within the AoR, which is essentially a user-defined "distance". Rather than being close (within the AoR) at the specific peak impact time, it has to be within the AoR during a user-specified time window. Requiring this at the exact peak impact time is essentially an equivalent, but more strict criterion than what we already have. Instead of implementing this as a post-processing step that filters out identified tracks based on a distance threshold, we would like to propose that we implement your suggestion in comment "R1C24", where we in the revised code automatically will choose the ETC track that is closest to the impact location at the impact time.

3. We prefer to use a rectangular AoR since this is easier and more efficient to work with programmatically, when subsetting and slicing many 2D fields. In many specific use cases (such storm surges in Denmark), there can be good arguments for not choosing an AoR that is symmetrical around the impact location (such as a circle with a certain radius). In this case, the geographical extent of the semi-enclosed Baltic Sea means that relevant ETCs can be located further to east and north of the impact location (Denmark) than to the west and south. That is why we choose an AoR that extends further north and east in this study. In the revised manuscripts, we will point the user to the exact parts of the code where the AoR is defined, and what the user needs to do to implement a custom, non-rectangular AoR if that is desired.

**R1C16** There is also a question of the wind direction the impact location is experiencing. Depending on where you are in relation to the storm center being

tracked you could be experiencing winds in opposite directions, which would directly impact the surge experienced. Adding such a criterion could help limit mis assigned storms.

It is correct that where you are relative to the storm's location will determine the wind direction you are locally experiencing. However, developing a criterion based on this is easier said than done, especially for complex coastal regions such as the Southern Baltic Sea with its semi-enclosed nature containing narrow entrances at the Danish straits and its many small islands. We imagine that this would require a lot of work by the users leading up to employing the algorithm. Users would need detailed information on the physical processes causing impacts, e.g. local, oceanographic knowledge of which wind directions a location is sensitive to. That would actually be something that storm tracking could help users figure out in the first place, so we would risk circular applications. (In fact, one of the applications that our research group intends to use this algorithm for in the future is to figure out which types of storms produce the impacts in different locations in complex coastal regions).

**R1C17** Figure 7. Why are there storm centers that are within one 350km pruning radius of one another? Shouldn't these have been pruned or connected in time already given the 600km threshold of HA connections in time. Numbers 13,60,27, 73, 35.. all seem like their connecting 2 points in very close proximity to one another.

This issue relates to how we select candidate points. In cases where there are multiple local minima with the exact same MSLP value at time $t$ within the pruning radius distance, we keep both points as potential candidates rather than making an arbitrary decision between one of them. When the HA/NN algorithm then has to connect candidate points through time there is a chance that the track ends prematurely, if the connection from $t-1$ to $t$ is to one of the points, but the continuation onwards from $t$ to $t+1$ continues from the other point. We let the BLOB step of the algorithm handle this issue by connecting these two fragmented tracks. We now realize that this is not explained well in the manuscript and will revise the text accordingly.

**R1C18** Table 2. I'm not convinced the optimal solution from HA is physically correct with regards to real world tracking, therefore I find it inappropriate to use it as ground truth to the nearest neighbor approach. How would this method compare to a small subset of hand drawn tracks?

We agree with the reviewer, that the optimal solution from HA, while guarenteeing a mathematically optimal solution for the t to t+1 correspondence problem, does not necessarily result in physically correct tracks with regards to real world tracking. We will re-write the language regarding optimality and ground truth, and instead frame our implementation of HA as a novel application to storm tracking, which we benchmark against a "standard" nearest neighbour approach. In the revised manuscript, we will present visual examples

of the tracking, including animations, so that readers can gain a better understanding of how the algorithm performs.

**R1C19** 283: This may be true in terms of minimizing distance summed across all tracks, but I don't see clear evidence of this method producing "More accurate tracks." This implies the shortest track is always the most accurate which seems like it wouldn't be true in all cases. One specific example is the case of a storm center splitting into 2, this tracking method would miss a potential storm track if one center was slightly further away at one time step, but was longer lasting overall. Wouldn't it be better to track both or perhaps even the longer lasting segment that was initially further away?

This relates to the previous comment (R1C18). By itself, the HA simply minimizes the summed distance across all tracks, but importantly it only chooses the "shortest track" connection for each individual time step. In the specific example of a track splitting into 2, we do in fact only end up with a single track, which is a limitation of the algorithm. However, the BLOB step of the algorithm will actually select the overall longest lasting of the two split tracks as the final storm track, not the shortest one. In the revised manuscript, we will make sure to explain this, and avoid the use of the word "most accurate", since this is only guaranteed from a mathematical optimization perspective.

**R1C20** Line 298: What was the nature of the one event not tied to a storm track?

The impact was a storm surge on the island of Bornholm in the Baltic Sea. The surge was based on easterly winds, which were caused by strong pressure (MSLP) gradients over the Baltic Sea. The gradients formed due to the interaction between a minor ETC located over Ukraine and a high-pressure system that develops over Southern Norway. Ideally, the algorithm should have identified the minor ETC over Ukraine. When we inspect the candidate locations and tracks from the algorithm, they show that the track is not inside the AoR for long enough to exceed the calibrated post-processing parameter that requires a minimum number of time steps inside the AoR. As we discuss in the manuscript, such cases show that a single, unique parameter set will never perfectly capture all ETC's. However, given that we for the revised manuscript will change several key parameter values, and implement additional steps in the algorithm, this ETC may not be missed in the revised implementation and manuscript. So, we will wait to see if this changes before we decide to make any changes to the manuscript here.

**R1C21** Line 315: Varying AoR by degrees lacks physical meaning as distance between lines of longitude will vary substantially depending on where you are in the domain. Why not use a kilometers-based approach?

While this is essentially true, it is also a function of the size of the domain

(variation within the study area) and location of the domain in the latitudinal direction (the effect of the relation between degrees and kilometres). For our study domain, the effect is small, but if the algorithm is used on other domains this could be a significant concern. We will therefore provide a significant discussion of this issue, and add the switch to a kilometres-based approach to the list of future improvements in section 5.3

**R1C22** Line 371: Alignment with the underlying MSLP field is very difficult to observe in the still images covering a wide time/space range. I think an animation in the supplementary materials would be more affective.

We will provide animations for example events in supplementary information, highlighting the underlying mslp field.

**R1C23** Line 375-376: These large jumps could be due to the large search radius of 600km, as this occurs while the storm center is still offshore but seems to be getting distracted by noisier MSLP minima along the coast, or perhaps the merging of another low-pressure system. Would choosing a smaller initial search radius and allow the BLOB method to connect the tracks after improve the result?

The case event referred to here, "2000-01-29" in Figure 11, has a vaugely defined pressure minima, with a large area of similar MSLP values. That causes the jumps, there is nothing for us to do during the "stitching". The track jumps because the minima in the vague field jump. This would have to be "fixed" by a new post-processing step that smoothes the tracks, which some other algorithms have decided to implement. Another fix would be to track areas rather than points in time, as e.g. implemented in the TempestExtremes tracking algorithm. However, we consider this outside the scope of our algorithm.

**R1C24** Figure 11 (Lines 377-378): how were orange and green tracks determined to be irrelevant? I think different post processing criteria could be applied to improve performance of the algorithm. Why not add a step that chooses the storm track closest to the impact location at the height of the event or something similar?

The orange and green tracks are not irrelevant. Here, our algorithm has identified three independent tracks that could the "culprit" of the impact, but the algorithm does not go further than this and it is up to the users to select between the blue, green and orange tracks, if they only want a single track. The different colours simply mean that the tracks were independent, not irrelevant or filtered out (only the red colour in our case examples means that a track has been filtered out by the post-processing steps).
We believe that the reviewer's suggestion of choosing the track that is closest to the impact location at the height of the event is a good one, and we have implemented this for the revised manuscript. If multiple locations are impacted

at the same time, we now associate each impact location the closest identified track.

Motivated by this comment, we have also decided to add a new step for cases where no tracks for an event survived the post-processing steps. In this case, we now search for relevant tracks that were filtered out to see if we can identify one. We will make sure to describe the new additions in the revised manuscript.

**R1C25** Line 384: how is the relationship between steep gradient and water level quantified? Is timing of the ETC and water level considered?

With this sentence, we simply mean that steep pressure gradients cause strong winds, which cause the storm surge. This was just a descriptive analysis of the event, and no quantitative analysis was done. We will highlight this in the manuscript.

**R1C26** Line 430: I agree that more work should be done to assess the interaction of multiple ETCs and I think some of the conclusions drawn about storms being "impact irrelevant" may be premature here. Since the manuscript is focused on impact on sea level, I think some of these steps should be addressed here. It should be relatively low effort to apply one or two post processing steps to the current work.

We agree with the reviewer, see response to R1C24.

**R1C27** 431-432: Wind speed could also be investigated to limit the filtering out of impactful storms.

We agree, with the caveat that filtering based on additional variables (here wind speed) would increase computational demand as discussed earlier. But we will add a discussion of this to the future improvements options, which the reviewer refers to here.

**R1C28** 449: As previously stated, I think suboptimal is poorly defined and the global optimal solution should not be used as benchmark in this manner.

We agree with the reviewer, and we will soften the wording.

**R1C29** 452-453: ETCs exist on timescales more than adequately resolved by the model being used here. One could even argue the high resolution is already unnecessary for resolving these large atmospheric features and could result in more confusion and runtime for algorithms as the number of small pressure perturbations increase. As resolution increases it is likely that model output would be down sampled spatially, much in the same way that the upscale gridding is performed here. I don't see the practicality of ETC tracking with higher resolution.

This issue relates to a comment made by the other reviewer. We therefore reply along the same lines for both of these inquiries.

In general, we fully acknowledge the points of both reviewers. We see two separate issues here: (1) whether higher resolution is unnecessary for resolving large atmospheric features, and (2) whether tracking is beneficial in higher resolution.

As a general rule, finer-scale models are able to better resolve not just atmospheric patterns themselves, but also processes and physics that in themselves lead to better results, also for storm tracks specifically (see e.g. Polichtchouk et al. (2025): "Effects of Atmosphere and Ocean Horizontal Model Resolution on Tropical Cyclone and Upper-Ocean Response Forecasts in Four Major Hurricanes". Monthly Weather Review). Moving to finer scales in geoscience has been the general trend, in line with advances in computational power, for decades. It is also likely the case that associated hazards (precipitation, wind fields, etc.) may be better resolved with higher resolution. The reviewer is likely right that there will not be major benefits to tracking ETCs in very high resolution grids. However, there may be benefits in terms of tracking other atmospheric objects, such as convective rainfall cells. We will therefore revise the manuscript text to highlight these points.